# Assessing Suitability of Sorghum to Alleviate Sub-Saharan Nutritional Deficiencies through the Nutritional Water Productivity Index in Semi-Arid Regions

**DOI:** 10.3390/foods10020385

**Published:** 2021-02-10

**Authors:** Sandile T. Hadebe, Albert T. Modi, Tafadzwanashe Mabhaudhi

**Affiliations:** 1Department of Plant Production, Soil Science and Agricultural Engineering, University of Limpopo, Private Bag Box X1106, Sovenga 0727, South Africa; 2Centre for Transformative Agricultural and Food Systems, School of Agricultural, Earth and Environmental Sciences, University of KwaZulu-Natal, Private Bag X01, Scottsville, Pietermaritzburg 3209, South Africa; modiat@ukzn.ac.za (A.T.M.); mabhaudhi@ukzn.ac.za (T.M.)

**Keywords:** nutritional security, food security, water scarcity

## Abstract

Lack of cereal nutritional water productivity (NWP) information disadvantages linkages of nutrition to water–food nexus as staple food crops in Sub-Saharan Africa (SSA). This study determined the suitability of sorghum (*Sorghum bicolor* L. Moench) genotypes to alleviate protein, Zn and Fe deficiency under water-scarce dryland conditions through evaluation of NWP. Sorghum genotypes (Macia, Ujiba, PAN8816, IsiZulu) NWP was quantified from three planting seasons for various sorghum seed nutrients under dryland semi-arid conditions. Seasons by genotypes interaction highly and significantly affected NWP_Starch, Ca, Cu, Fe_, and significantly affected NWP_Mg, K, Na, P, Zn_. Genotypic variations highly and significantly affected sorghum NWP_Protein, Mn_. Macia exhibited statistically superior NWP_protein_ (13.2–14.6 kg·m^−3^) and NWP_Zn_ (2.0–2.6 g·m^−3^) compared to other tested genotypes, while Macia NWP_Fe_ (2.6–2.7 g·m^−3^) was considerably inferior to that of Ujiba and IsiZulu landraces under increased water scarcity. Excellent overall NWP_protein, Fe and Zn_ under water scarcity make Macia a well-rounded genotype suitable to alleviating food and nutritional insecurity challenges in semi-arid SSA; however, landraces are viable alternatives with limited NWP_protein and Zn_ penalty under water-limited conditions. These results underline genotype selection as a vital tool in improving “nutrition per drop” in semi-arid regions.

## 1. Introduction

Sub-Saharan Africa (SSA) faces twin challenges of food and nutritional insecurity [1], which are particularly higher in rural, resource-poor households [2] where approximately 85% of the population depends on small-scale, rainfed agriculture for their livelihoods [3]. Crop production and yields in SSA are negatively impacted by low, irregular and erratic rainfall where 43% of the region is arid and semi-arid, and approximately 95% of agriculture being rainfed [4]. The occurrence of droughts exacerbates water scarcity problems encountered in the region and necessitates a shift to drought-tolerant alternatives to achieve reasonable yields under water scarcity [5]. Macro- (e.g., proteins) and micronutrient (e.g., zinc and iron) deficiencies or “hidden hunger” are prevalent in SSA and affect most low-income households due to poor or inadequate diets based on starchy staple food crops [6,7]. The issue of malnutrition is further exacerbated by low crop yields under water scarcity, which reduces the food supply from which nutrition is derived [8].

To address food and nutritional insecurity, the focus needs to be placed not only on drought-tolerant crops that can produce reasonable yields under water-limited conditions but on nutrient-dense crops under water-scarce conditions. To simultaneously address quantity (food produced per unit of water used) and quality (nutrition per drop) components of food security [9] under water scarcity in SSA, an index that combines crop yields, water availability, and crop nutrition is fitting [10]. Nutritional water productivity (NWP) is an emerging concept that combines information of nutritional value with that of crop water productivity, making it a useful index for evaluating the impacts of agriculture on food and nutrition security [11]. The result is an index that includes nutritional value-based output per unit of water used [10]. Under water-scarce conditions, crop and irrigation experts have historically focused on producing “more crop per unit of water used” (water productivity), whereas nutritionist research has focused on meeting the daily recommended human nutrition requirements [12]. This has led to a lack of attention to the inter-linkages between food/crop production, human health and nutrition. The lack of linkages between food production and human nutrition has often led to agricultural interventions being disconnected from issues of human health and nutrition [11].

The application benefit to measuring crop NWP values is the ease of selecting crops that maximize the nutrition of key nutrients per drop of water and ease of selecting crops with sustained nutrient density under water-scarce environments reminiscent of crop growing conditions in semi-arid SSA. Research on NWP of drought-tolerant, nutrient-dense crops has been scarce; where present, it has focused on the contribution of leafy vegetables [1,12] and legumes [13]. This is despite the fact that cereals are the main staple food crops in SSA, and sorghum (second most produced and consumed cereal) is known for its drought-tolerance qualities [5], producing reasonable yields under water scarcity. Sorghum contains up to 21.1% protein in the seed and is rich in iron and zinc, which are deficient nutrients in SSA diets [14,15], making it a candidate crop to simultaneously address food and nutrition insecurity in the region. Sorghum’s nutritional composition is known to be influenced by genotypic differences and water scarcity among many crops and environmental factors [16,17]. To this end, the lack of NWP values for sorghum is a disadvantage, especially in understanding how NWP of sorghum is impacted by water scarcity. The primary aim of this study was, therefore, to determine NWP values of various sorghum genotypes under different water availability conditions in rainfed agriculture. Secondary to that, determine the impact of water scarcity on sorghum NWP of protein, zinc and iron as key deficient nutrients in SSA diets and suitability of sorghum to alleviate said deficiencies under water-limited conditions.

## 2. Materials and Methods

### 2.1. Plant Material

Four genotypes of sorghum were used, namely PAN8816, Macia, Ujiba and IsiZulu. These genotypes reflect the range of germplasm typically used by farmers for sorghum production in Southern Africa. PAN8816 is a semi-dwarf, bronze-grained, medium- to late-maturing, low-tannin hybrid, which was sourced from Pannar Seeds^®^; and represents hybrids as a popular seed choice by commercial sorghum farmers. Macia is an early to medium-maturing, semi-dwarf (1.3–1.5 m tall with thick stem), low-tannin open-pollinated variety developed by the International Crops Research Institute for the Semi-Arid Tropics (ICRISAT) breeding program for improved drought-tolerance, protein content and yield potential (3–6 t·ha^−1^). Macia represents a popular seed choice by both commercial and small-scale farmers and is grown across SSA. Ujiba is a reddish-brown seeded, tall-growing (>1.5 m), high-tannin landrace genotype sourced locally from smallholder farmers in Tugela Ferry (28°44′ S, 30°27′ E). IsiZulu is a dark-brown seeded, tall-growing (>1.5 m), high-tannin landrace genotype sourced locally from smallholder farmers at Nkandla (28°50′ S, 31°06′ E). Both landraces were sourced from KwaZulu-Natal province in South Africa and represent an affordable source of planting materials popular among small-scale farmers in Southern Africa.

### 2.2. Site Description

Field trials were planted at Ukulinga Research Farm (30°24′ S, 29°24′ E, 805 m above sea level) in Mkhondeni, Pietermaritzburg, KwaZulu-Natal province. Ukulinga is classified as a semi-arid environment, and soils are classified as Arcadia form, Lonehill family [18] by South African classification. Rain falls mostly in summer between September and April, and rainfall distribution varies annually, with the bulk of the rain falling in November, December and early January. Occasionally, light-to-moderate frost occurs in winter (May–July).

### 2.3. Experimental Design

The field trial experimental design is as detailed in [19]. The experiment was a split-plot design with planting seasons (first, second and third) as the main factor and genotypes (PAN8816, Macia, IsiZulu, Ujiba) as the subfactor laid out in randomized complete blocks with three replications. To achieve three planting seasons, sorghum was planted over three planting dates (early, optimal and late), which were on 3 November 2014, 17 November 2014, and 26 January 2015, respectively. Planting dates were designed to mimic variable water availability scenarios under rainfed agriculture (Table 1). The early planting date was reflective of the onset of rainfall at Ukulinga and mimicked a scenario of low water availability occurring mainly during the sowing and seedling emergence stage of sorghum growth. The selection of this planting date was based on when the onset of seasonal rainfall occurred. The optimal planting date is within planting time recommended by the Department of Agriculture, Fisheries and Forestry [20] for achieving optimal yields under rainfed agriculture. The late planting date mimicked low water availability occurrence, mainly during the reproductive (flowering and grain filling) stages of sorghum. Selection of the late planting date was based on two reasons: (i) sufficient rainfall for sorghum genotypes to reach physiological maturity and (ii) avoidance of cold stress and frost to allow sorghum to mature before the onset of the winter season. Rainfall received at each planting date is indicated in Table 1, where recommended rainfall distribution is represented by the crop coefficient and compared to actual rainfall received at different sorghum growing stages during three growing seasons. The main plot size (a single planting season trial) was 391.5 m^2^, and the sub-plot size (each genotype) was 6 m × 4.5 m (27 m^2^), with 1 m spacing between plots. Inter-row spacing was 0.75 m with 0.30 m intra-row spacing, amounting to 44,444 plants per hectare.

### 2.4. Agronomic Practices

Soil samples were collected using a soil auger from the top 30 cm soil layer and analyzed for fertility before land preparation. Before planting, fallow land was mechanically plowed, disked and rotovated using a tractor to prepare a planting seedbed. A glyphosate (Round-up) pre-emergence herbicide (10 mL per 1 L of water) was applied to control pre-emergence weeds 2 weeks before planting. A 120 kg·ha^−1^ application of nitrogen is recommended to achieve a 6 kg·ha^−1^ sorghum yield potential [21]. From soil analysis, a deficit of fertilizer requirements was applied using Gromor accelerator (30 g·kg^−1^ N, 15 g·kg^−1^ P and 15 g·kg^−1^ K), a slow-releasing organic fertilizer, 14 days after sowing (DAS) for each planting date. To supply deficit fertilizer requirements, 48 kg·ha^−1^ of fertilizer were applied for the first and second season, while 51 kg·ha^−1^ was applied for the third growing season. Planting rows were opened using hand hoes 3–5 cm deep, and seeds were hand-sown into the ground. At crop establishment (14 DAS), seedlings were thinned to the required spacing. Scouting for pests and diseases was conducted weekly, leading to the application of Cypermethrin^®^ (15 mL per 10 L knapsack) four weeks after planting to control insect pests. Weeding was conducted using hand-hoes at frequent intervals.

### 2.5. Atmospheric Data and Soil Characterization

Daily climatic data were obtained from an on-station (within 100 m radius from field experiments) automatic weather station, courtesy of the Agricultural Research Council—Institute for Soil, Climate and Water (ARC–ISCW). Daily minimum and maximum temperature, rainfall and reference evapotranspiration were collected weather station records (Figure 1).

Soil physical and hydraulic properties were obtained from a previous classification and characterization of experimental site soils by [19]. These included volumetric water content at field capacity, permanent wilting point, saturation, as well as saturated hydraulic conductivity and soil depth (Table 2).

### 2.6. Grain Yield, Water Use and Crop Water Productivity

Sorghum grain was collected from panicles at harvest maturity from three plants per replicate, and grain yield (Y) was calculated as a function of panicle mass over the area. A grain moisture meter (Nunes Instruments, Coimbatore, Tamil Nadu, India) was used to ensure that grains were below 12.5% moisture content. Soil water content was measured weekly from sowing to physiological maturity using a PR2/6 profile probe (Delta-T, Cambridge, UK). Weekly measurements of soil water content (SWC) were used to compute the soil water balance for rainfed trials [8] as follows:ETa = P ± ΔSWC(1)
where: ETa = crop water use (actual evapotranspiration), P = rainfall, and ΔSWC = change in soil water content. Rainfall received was observed from sowing to physiological maturity.

Actual field evapotranspiration (mm) obtained was used to calculate water productivity (WP) together with sorghum grain yield using the following equation [8]:Water productivity = Y/ETa(2)

### 2.7. Nutritional Composition of Sorghum Grain

Grain samples were milled to obtain fine grain powder that was used in the assessment of nutritional composition. Nutritional guidelines consider energy, total proteins, lipids, vitamins, minerals and amino acids [10] to assess nutrition. In this study, the nutritional assessment was conducted for starch, total proteins, Mg, Ca, K, Na, P, Zn, Cu, Mn and Fe in order to encompass major sorghum nutrients while covering key deficient nutrients in SSA population diets.

To determine starch and protein content, harvested material was freeze-dried at −60 °C for 72 h, and thereafter ground through a 1 mm screen of a mill hummer. Chemical analysis was conducted following the Association of Official Analytical Chemists standard procedures [19]. Dry matter (DM) was determined by drying samples in a fanned oven at 100 °C for 24 h. Nitrogen (N) was determined by the micro-Kjeldahl method, and crude protein (CP) will be calculated as N × 6.25. The ether extract was determined according to the Soxhlet procedure (AOAC 920.39). Ash was determined by igniting fiber samples in a furnace at 550 °C overnight (AOAC 942.05). The carbohydrate content was determined by difference, the addition of all the percentages of moisture, fat, crude protein, ash, and crude fiber subtracted from 100%. This resulted in the amount of nitrogen-free extract, which is the carbohydrate.

To determine the mineral composition of sorghum grain, the dry-ashing (DA) technique was used. The DA technique is probably the most used approach for the mineralization of organic-based samples. The mineral composition was determined for Ca, Co, Fe, K, Mg, Mn, Na, P, and Zn. An aliquot of 25 mL of each sample was placed in porcelain crucibles. To avoid cross-contamination between the samples, single used plastic tools were used to transfer samples. Thereafter, samples were placed in a low-temperature oven (50 °C) and heated overnight. Following this, crucibles with residues obtained after vaporization of water and most organic compounds were then introduced in a high-temperature muffle furnace and ashed at 450 °C. The temperature in the muffle oven was increased at a rate of about 50 °C per hour and maintained at 450 °C for 18 to 24 h. Thereafter, samples were cooled, and residues treated with nitric acid with gentle warming on a hot plate. Following this, all samples were transferred to the muffle furnace for 24 h under similar conditions as before. White ashes obtained were dissolved in a beaker with 20 mL 5% (*v/v*) nitric acid. After the solution, the content was transferred to a 25 mL volumetric flask by rinsing with 5% *v/v* nitric acid and adding the washing liquids until the mark, and the results were expressed on a % *w/w* basis.

### 2.8. Nutritional Water Productivity

A definition of nutritional water productivity (NWP) [10] was used to calculate NWP as follows:NWP = WP × NC(3)
where WP is water productivity calculated from field measured grain yield (Y, in kg·ha^−1^) and crop water use (ETa, in mm); and NC is the nutritional content per unit mass of the product (nutrition mass unit per 100 g of seed). NWP was expressed in units of nutrients per unit of water (e.g., kg·m^−3^).

### 2.9. Data Analyses

Data were subjected to analysis of variance (ANOVA) using GenStat^®^ Version 20 (VSN International, Hemel Hempstead, UK) using a split-plot design to observe the difference between treatments. Means were separated using the least significant differences (LSD) at a probability level of 95%. Multiple mean comparisons were conducted using Bonferonni test, where letters were assigned to individual means. Means that shared a common letter(s) were not significantly different from each other, while means not sharing a similar letter(s) were considered statistically different.

## 3. Results

Table 2 presents nutrient density of various sorghum genotypes as affected by water scarcity under dryland conditions, including the nutrient density of key SSA deficient nutrients such as proteins, Fe and Zn. Macia had high protein nutrient density across planting dates (128–147 g·kg^−1^) relative to PAN8816 (118–138 g·kg^−1^), Ujiba (104–115 g·kg^−1^), and IsiZulu (93–117 g·kg^−1^). Iron nutrient density was considerably higher in IsiZulu and Ujiba landraces (462–707 mg·kg^−1^) under water scarcity, whereas iron density was maintained for improved PAN8816 and Macia genotypes under water scarcity. Sorghum genotypes considerably differed in Zn density where Macia and Ujiba exhibited high Zn content in comparison to IsiZulu throughout planting seasons (Table 3). The interaction of seasons by genotypes significantly influenced (*p* < 0.05) sorghum water productivity. Ujiba and IsiZulu water productivity were higher under severe water scarcity in the third season (0.84–0.97 kg·m^−3^) compared to other higher rainfall planting seasons (Table 3), even though the highlighted improvement was not statistically significant. Water productivity of Macia and PAN8816 improved sorghum genotypes decreased (0.76–0.92 kg·m^−3^) with low rainfall quantity in the third season, where again the highlighted decrease was not statistically significant. Relatively higher water productivity values may compound increases in NWP values due to a higher multiplier effect, even when the differences are statistically insignificant.

Results suggest that sorghum can provide up to a quarter (17–24%) of the protein recommended daily allowance (RDA) for human consumption, with relatively lower RDA contribution by landraces compared to improved genotypes (Table 4). Tested genotypes provided nearly 3 times Zn (274–293%) and Fe (219–344%) RDA for human consumption, with considerably increased Fe RDA contribution under water scarcity (Table 4).

The interaction of seasons by genotypes highly and significantly (*p* < 0.001) affected NWP_Starch, Ca, Cu, Fe_, and significantly (*p* < 0.05) affected NWP_Mg, K, Na, P, Zn_. Differences in NWP_Protein, Mn_ were insignificantly influenced by the main study interaction. However, genotypic variations highly and significantly affected sorghum NWP_Protein_ and NWP_Mn_ (Table 5). Nutritional composition and water productivity differences (Table 2) largely informed significant differences in NWP observed between planting dates and genotypes. Under water-scarce conditions in the third season, sorghum genotypes maintained statistically similar NWP_Protein_ (10.7 kg·m^−3^) to when water availability was considerably (11.2 kg·m^−3^) in the second season. Genotype choice statistically and significantly affected NWP_Protein_, where Macia had superior NWP_Protein_ (13.2–14.6 kg·m^−3^) compared to landrace sorghum genotypes during the first and second seasons. Under increased water scarcity in the third season, all genotypes exhibited statistically similar NWP_Protein_.

Iron NWP significantly increased (3.3 g·m^−3^) under increased water scarcity at late planting compared to higher rainfall at optimal (2.6 g·m^−3^) and early (2.2 g·m^−3^) planting dates. Ujiba (5.1 g·m^−3^) and IsiZulu (3.5 g·m^−3^) landraces exhibited statistically superior NWP_Fe_ under increased water scarcity in comparison to Macia and PAN8816 improved genotypes. In the current study, sorghum exhibited genotypic by seasonal differences in NWP_Zn_ under different water scarcity levels. Macia, PAN8816 and Ujiba consistently maintained high NWP_Zn_ (1.8–2.6 g·m^−3^) under all tested semi-arid rainfall levels (Table 5). Under relatively high rainfall levels in the first and second season, Macia (2.5–2.6 g·m^−3^) statistically outperformed IsiZulu (1.2–1.6 g·m^−3^) NWP_Zn_, while no significant differences were observed between genotypes under high water scarcity in the third season.

## 4. Discussion

The study set out to determine the NWP of selected sorghum genotypes commonly cultivated in SSA under different semi-arid rainfall availability and variability levels to gain insight into how water availability and variability affect nutrition in sorghum. Macro (fat, carbohydrate and protein) and micronutrients (vitamins and minerals) deficiencies have a considerable negative impact on human health and development in SSA [6]. However, this study places emphasis on how water scarcity affects nutrition per drop of deficient nutrients in SSA communities such as proteins, Zn and Fe. Water scarcity in this study is mainly referred to with respect to late planting in the third planting season, where water received (267 mm) was considerably lower than the sorghum crop water requirements for optimal growth and yields of 450–650 mm [25]. However, less than optimal rainfall was also received for early (418 mm) and optimal (401 mm) planting dates, highlighting the semi-arid nature of the study area. Accumulation of nutrients occurs in sorghum during grain filling and is regulated by enzymes and associated plant hormones [26]. Water forms an integral part of nutrient accumulation and translocation biochemical pathways in sorghum, and the lack of this critical compound can affect the nutritional composition in sorghum grain. Water scarcity can reduce nutrient uptake through various methods, including the reduction of nutrient supply through mineralization and by affecting the kinetics of nutrient uptake by roots [27].

Nutritional composition and water productivity differences (Table 3) largely informed highly significant (*p* < 0.001) and significant (*p* < 0.05) differences in NWP_Starch, Ca, Cu, K, Na, Fe, Mg, P and Zn_ observed in the interaction of seasons by genotypes (Table 5). Statistical differences in NWP results were therefore attributed to different water scarcity and rainfall distribution levels at different planting seasons, as water availability and variability strongly influenced nutritional composition, crop yield, and water use of sorghum genotypes. Results are unsurprising given that water forms an integral part of most biochemical pathways of plants, thereby has direct involvement in determining crop WP, grain nutritional composition and inadvertently crop NWP.

### 4.1. Protein NWP

Proteins from the second major component of sorghum grains after starch and their content in seeds is affected by both genetic and environmental factors. The dietary protein requirements vary between 8 and 67 g·day^−1^, with infants under 12 months having the highest dietary protein requirements. Of the four tested genotypes, protein composition varied between 9 and 15 g·100 g^−1^ of sorghum seeds, where protein content was consistently higher in Macia and PAN8816 in comparison to landrace sorghum genotypes. Study results suggest that sorghum can provide up to a quarter of the protein RDA for human consumption, with relatively lower RDA contribution by landraces compared to improved genotypes. The relatively higher protein content in the Macia open-pollinated variety (OPV) and PAN8816 hybrid are due to breeding and biofortification efforts for high protein content in improved genotypes [28]. Sorghum landraces are mainly grown and consumed by resource-poor farmers in arid and semi-arid regions as a staple food crop without selection for improved protein content, which makes the selection of landrace cultivars with high ‘protein nutrition per drop’ critical for this food and nutritionally vulnerable group. Ujiba and IsiZulu landraces exhibited statistically similar water productivity to improved genotypes under water scarcity at late planting; however, relatively lower protein content in landraces reduced protein NWP compared to Macia open-pollinated variety, which has been bred particularly for high protein content under water-limited conditions. Sorghum proteins are generally less digestible than other cereals due to extensive polymerization of kafirins upon cooking, together with the presence of tannins in certain sorghum lines, which reduces the bioavailability of seed proteins [17]. Lower protein content together with known high tannin content in Ujiba and IsiZulu landraces suggests that low-tannin, high protein improved genotypes such as Macia should be recommended for protein deficient populations in place of landraces.

Differences in NWP_Protein_ were insignificantly influenced by the main study interaction of seasons by genotypes; however, genotypic variations highly and significantly influenced differences in NWP_Protein_ (Table 5). Genotype choice statistically and significantly affected NWP_Protein_, where Macia had superior NWP_Protein_ (13.2–14.6 kg·m^−3^) compared to Ujiba and IsiZulu landraces in the first and second planting season. In this study, under water-scarce conditions in the third planting season, sorghum maintained statistically similar NWP_Protein_ to the optimal planting date where water availability was significantly higher. In sorghum, grain protein content is expected to increase with increasing water stress [29], which technically should increase protein NWP under water stress. The results of this study, therefore, differ from the expectation of increased NWP_Protein_ under increasing water stress. Worth noting is that studies by [29] were conducted under irrigated conditions, the main difference being that water levels and frequencies were controlled in the cited literature compared to rainfed production in this study. Maintained high NWP_Protein_ under increased water stress suggests that sorghum retains high protein density under water scarcity, making it particularly suitable to addressing protein deficiency in rural and resource-poor households of semi- and arid regions of SSA. In the event that improved, drought-tolerant genotypes (e.g., Macia) are not a viable production option to farmers, landraces can be produced with a minimal penalty on NWP_Protein_.

### 4.2. Iron NWP

Of the four micronutrients (iron, zinc, iodine and vitamin A) identified by the Committee on Micronutrient Deficiencies in 1998 as limiting in developing countries, two (Fe and Zn) were tested in this study. Study results suggest that increasing water scarcity significantly increases Fe content in sorghum, and landraces (Ujiba and IsiZulu) tend to have higher Fe content compared to improved genotypes (Macia and Ujiba) (Table 3). Studies on sorghum nutrition agree with research findings that sorghum is an excellent source of iron and that landraces tend to possess higher Fe content than improved genotypes [16]. Ujiba landrace contained more than five times the iron RDA under water-scarce conditions when sorghum was planted late, which was significantly higher than all tested genotypes (Table 4). The challenge with using landraces in experimentation is that the individuals tend to be highly heterogeneous, and results will vary greatly even after replicating [30], therefore results from this study need further verification.

Iron NWP significantly increased-under-increased water scarcity in the third season compared to considerably higher rainfall in the first and second planting season, with landraces exhibiting statistically superior NWP_Fe_ in the third season compared to improved genotypes. Iron levels have been reported to reduce under drought stress in many plant species [31,32], which could lead to reduced NWP_Fe_ in crops. Contrary to literature expectations, research findings suggest that NWP_Fe_ significantly increases with increasing water scarcity in sorghum genotypes. Increased NWP_Fe_ with water scarcity was explained by high average Fe content obtained at the late planting date relative to maintained average water productivity. Ujiba and IsiZulu landraces exhibited significantly higher NWP_Fe_ compared to improved PAN8816 and Macia genotypes, which was attributed to increased WP and Fe content under water scarcity at late planting. These findings suggest that breeding for increased Fe under water scarcity is possible in sorghum through gene combination with landraces and selection for increased Fe under water scarcity. Another alternative is biofortification [17] of drought-tolerant improved sorghum genotypes (e.g., Macia) to increase Fe content. The findings of high NWP_Fe_ under increased water scarcity suggest that sorghum retains or increases iron density under water scarcity, making it a suitable option to combat Fe deficiency in food and nutritionally vulnerable populations of semi- and arid SSA. Furthermore, findings suggest that varietal selection for landraces can be a vital tool for increasing Fe nutrition per drop in nutritionally vulnerable populations of semi- and arid areas, where Ujiba can be championed as a potential candidate for mitigating Fe deficiency.

### 4.3. Zinc NWP

Sorghum exhibited genotype by seasonal differences in NWP_Zn_ under different water scarcity levels. Macia, PAN8816 and Ujiba consistently maintained high NWP_Zn_ under all tested semi-arid rainfall levels. When rainfall was considerably high in the first and second seasons, Macia statistically outperformed IsiZulu NWP_Zn_, while no significant differences were observed between genotypes under high water scarcity in the third season. This suggests that where improved cultivars are not a viable option to nutritionally vulnerable households, landraces such as Ujiba can be cultivated to meet Zn nutritional requirements in semi- and arid areas. Sustained high NWP_Zn_ under increasing water scarcity in Macia showcases the benefits of deliberate breeding for improved drought and nutrition [33]. Ref. [34] reported sustained Zn content in chamomile in spite of increasing water scarcity, which agrees with NWP_Zn_ findings in PAN8816, Macia and Ujiba. Elsewhere in castor plants, decreasing Zn content with increasing water scarcity has been reported [29], which contradicts study findings. Another school of thought observed in common marigold [35] suggests that Zn in drought tolerant crops/genotypes should increase under water scarcity since high Zn content is associated with improved drought tolerance. Further research is required to explain genotype-specific NWP_Zn_ responses to water scarcity in sorghum due to lack of literature, to compare how water scarcity influences seed Zn content in sorghum. The high and sustained NWP_Zn_ observed in sorghum genotypes may be explained by the role of Zn in seedling drought-tolerance [34,36], where high Zn content under low water availability increases seedling water productivity. This theory may also explain decreasing Zn content with increasing water scarcity observed in relatively drought susceptible PAN8816 hybrid sorghum. However, this needs further verification. Both Macia and Ujiba can thereby be recommended to combat Zn deficiency in food and nutritionally vulnerable populations of semi- and arid SSA.

### 4.4. Discussion Summary

Study findings suggest that Macia open-pollinated variety is particularly suited to alleviate protein and Zn nutritional security in semi-arid regions due to superior NWP_protein and Zn_ under increased water scarcity. Ujiba and IsiZulu landraces are recommended to alleviate Fe deficiency in semi-arid regions because of superior NWP_Fe_ compared to improved genotypes under increased water scarcity at late planting. Worth noting, Macia and PAN8816 improved genotypes met and exceeded Fe nutritional requirements under water scarcity, which qualifies these genotypes for Zn deficiency alleviation in semi-arid regions. In comparison to other tested genotypes, excellent overall protein, Fe and Zn NWP under water scarcity make Macia an overall well-rounded genotype suitable to alleviating food and nutritional insecurity challenges in semi-arid SSA. An added advantage of Macia over other tested genotypes is wider release and adoption of the genotype in SSA countries [37], which makes Macia cultivar easily available for selection by farmers when producing sorghum. Where high initial costs of OPV seed dissuade financially constrained farmers from growing Macia, landrace genotypes are recommended for production with a limited NWP_protein and Zn_ penalty under water-limited conditions.

## 5. Conclusions

This study determined the suitability of sorghum genotypes to alleviate protein, Zn and Fe deficiency in SSA under water scarcity in dryland agriculture through evaluation of nutritional water productivity and other associated parameters. Tested genotypes provided up to a quarter of protein RDA and fully met and exceeded Zn and Fe RDA. Of the tested genotypes, Macia was overall most suited to meet protein, Fe and Zn nutritional requirements due to statistically superior NWP_protein and Zn_ under increased water scarcity. Furthermore, Macia met and exceeded Fe RDA with excellent NWP_Fe_. However, landrace genotypes are recommended to meet Fe requirements under increased water scarcity in semi-arid environments due to superior NWP_Fe_, particularly under increased water scarcity. This study recommends that Macia OPV be further bred for increased Fe content through conventional breeding approaches and biofortification efforts to be a more well-rounded solution to protein, Fe and Zn deficiencies in SSA. Sorghum NWP values determined in this study were under dryland, suboptimal (<450 mm seasonal rainfall) conditions; there is a need to determine NWP values under variable water levels supplied under irrigation to further understand the effect of water quantity, frequency and stage of application on sorghum NWP. Experimental data from such trials could further be used to extend existing crop water productivity models to include modeling NWP of various crops and genotypes to develop NWP crop/genotype by environment predictive decision support tool.

## Figures and Tables

**Figure 1 foods-10-00385-f001:**
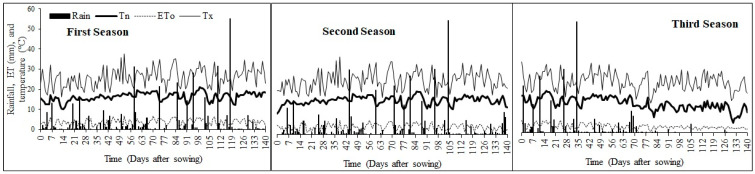
Daily rainfall (rain), reference evapotranspiration (ET0), minimum (Tn) and maximum temperatures at Ukulinga experimental site during three sorghum planting seasons.

**Table 1 foods-10-00385-t001:** Rainfall received at three key developmental stages when planted in three planting seasons (at three planting dates).

Season	Total Rainfall (mm)	Rainfall Received during Each Stage			Comments
		Initial	Development	Midseason	
First (early planting)	418	80	151	187	Low soil moisture during crop emergence and early vegetative stage. Rainfall distribution regular afterward.
Second (optimal planting)	401	79	173	179	Low soil moisture at sowing, thereafter regular recharge of soil moisture from rainfall. Sorghum was planted at recommended planting date.
Third (late planting)	267	204	41	22	Irregular rainfall distribution, increasingly low soil moisture and low, irregular rainfall after flowering.
Crop coefficient (Kc)		0.45	0.83	1.18	

Initial stage = period from sowing until the crop covers about 10% of the ground (crop establishment); development stage = period from crop establishment until 70–80% crop canopy ground cover; midseason stage = period from the end of development stage until crop physiological maturity.

**Table 2 foods-10-00385-t002:** Soil physical and hydraulic properties of Ukulinga experimental site.

Soil Taxonomy	Textural Class	Clay Content (%)	Bulk Density (g·m^−3^)	Field Capacity (mm·m^−1^)	Permanent Wilting Point (mm·m^−1^)	Saturation (mm·m^−1^)	Soil Profile Depth (m)	Saturated Hydraulic Conductivity (mm·day^−1^)
Vertisols	Clay loam	±29	1.2	406	230	481	0.6	25

**Table 3 foods-10-00385-t003:** Water productivity (WP) and nutrient density of selected sorghum genotypes at different planting dates.

Season	Genotype	WP	Starch	Protein	Ca	Mg	K	Na	P	Zn	Cu	Mn	Fe
		kg·m^−3^	–––––––––––––––––––––––––g·kg^−1^–––––––––––––––––––––––––––––							–––––––––––––––mg·kg^−1^–––––––––––––––			
First	PAN8816	0.98 ^abc^	420	118	0.07	0.13	0.35	0.002	41	270	16	136	243
	Macia	1.10 ^bc^	340	128	0.11	0.14	0.44	0.002	57	322	38	193	354
	Ujiba	0.68 ^ab^	320	115	0.09	0.19	0.41	0.002	39	287	10	164	349
	IsiZulu	0.53 ^a^	320	92	0.09	0.16	0.37	0.002	26	162	16	113	194
Mean		0.83	350	114	0.09	0.16	0.39	0.002	41	260	20	152	285
Second	PAN8816	0.94 ^abc^	380	138	0.09	0.13	0.35	0.002	42	271	16	163	297
	Macia	1.16 ^c^	350	126	0.11	0.14	0.42	0.002	60	346	52	209	346
	Ujiba	0.73 ^ab^	400	118	0.09	0.18	0.55	0.002	47	294	14	158	340
	IsiZulu	0.71 ^ab^	350	117	0.09	0.16	0.39	0.002	37	223	24	156	425
Mean		0.92	370	125	0.10	0.15	0.43	0.002	46	284	27	172	352
Third	PAN8816	0.76 ^abc^	450	126	0.13	0.15	0.42	0.002	40	250	25	159	294
	Macia	0.93 ^abc^	360	147	0.17	0.17	0.42	0.002	51	271	32	190	325
	Ujiba	0.97 ^abc^	420	104	0.18	0.17	0.39	0.002	51	325	24	206	707
	IsiZulu	0.84 ^abc^	410	113	0.15	0.18	0.47	0.002	45	267	24	171	462
Mean		0.87	410	122	0.16	0.17	0.43	0.002	47	278	26	181	447
Coefficient of variation (%)		14.4											
*p*-value (S × G)		0.009											
*p*-value (genotype (G))		<0.001											
*p*-value (season (S))		0.710											

^abc^ = means sharing a common letter(s) are not significantly different from each other, while means not sharing a similar letter(s) are statistically different.

**Table 4 foods-10-00385-t004:** Protein, zinc and iron contribution to the required dietary allowance (RDA) of different sorghum genotypes grown at different planting dates.

Season	Genotype	Protein			Zn			Fe		
		g·100 g^−1^	^1^ RDA (g·day^−1^)	% RDA	mg·100 g^−1^	^2^ RDA (mg·day^−1^)	% RDA	mg·100 g^−1^	^3^ RDA (mg·day^−1^)	% RDA
First	PAN8816	12		19	27		285	24		256
	Macia	13		21	32		339	35		272
	Ujiba	12		19	29		302	35		268
	IsiZulu	9		15	16		171	19		150
Mean		11	62	18	26	10	274	29	13	219
Second	PAN8816	14		22	27		285	30		228
	Macia	13		20	35		365	35		266
	Ujiba	12		19	29		309	34		261
	IsiZulu	12		19	22		235	43		327
Mean		13	62	20	28	10	298	35	13	271
Third	PAN8816	13		20	25		263	29		226
	Macia	15		24	27		286	33		250
	Ujiba	10		17	32		342	71		543
	IsiZulu	11		18	26		281	46		355
Mean		12	62	20	28	10	293	45	13	344

^1^ Average of the highest requirements from male (67 g·day^−1^) and female (57 g·day^−1^) 65 year and older [22]. ^2^ Average of adult males (11 mg·day^−1^) and females (8 mg·day^−1^) [23]. ^3^ Average of the adult males (11 mg·day^−1^) and females (15 mg·day^−1^) [24].

**Table 5 foods-10-00385-t005:** Nutritional water productivity of selected sorghum genotypes at different planting dates.

Season	Genotype	Starch	Protein	Ca	Mg	K	Na	P	Zn	Mn	Fe	Cu
		––––––kg·m^−3^–––––		–––––––––––––––––––––––––––––––––––––g·m^−3^––––––––––––––––––––––––––––––––––								–mg·m^−3^–
First	PAN8816	41.0 ^c^	11.6 ^bcd^	6.8 ^ab^	12.7 ^ab^	34.2 ^abcd^	0.20 ^bc^	32.2 ^abc^	2.1 ^abc^	1.1 ^ab^	1.9 ^ab^	127.0 ^ab^
	Macia	37.2 ^bc^	14.0 ^cd^	11.9 ^cd^	15.7 ^b^	48.1 ^cd^	0.22 ^bc^	43.0 ^c^	2.5 ^bc^	1.4 ^b^	2.7 ^abc^	276.4 ^cd^
	Ujiba	21.8 ^ab^	7.9 ^ab^	6.2 ^ab^	13.0 ^ab^	28.0 ^ab^	0.14 ^ab^	28.7 ^abc^	2.1 ^abc^	1.2 ^ab^	2.6 ^abc^	75.3 ^a^
	IsiZulu	17.2 ^a^	4.9 ^a^	4.8 ^a^	8.5 ^a^	19.7 ^a^	0.11 ^a^	18.6 ^a^	1.2 ^a^	0.8 ^a^	1.4 ^a^	117.0 ^a^
Mean		30.1	9.9	7.7	12.4	33.4	0.17	31.4	2.0	1.2	2.2	155.6
Second	PAN8816	35.7 ^bc^	13.0 ^bcd^	8.5 ^abc^	12.2 ^ab^	32.9 ^abcd^	0.19 ^abc^	32.0 ^abc^	2.1 ^abc^	1.2 ^ab^	2.3 ^abc^	122.2 ^ab^
	Macia	41.0 ^c^	14.6 ^d^	12.7 ^cde^	16.5 ^b^	49.6 ^d^	0.23 ^c^	46.6 ^c^	2.6 ^c^	1.6 ^b^	2.6 ^abc^	372.1 ^d^
	Ujiba	29.1 ^abc^	8.6 ^abc^	6.6 ^ab^	13.1 ^ab^	40.0 ^bcd^	0.15 ^ab^	32.8 ^abc^	2.1 ^abc^	1.1 ^ab^	2.4 ^abc^	94.8 ^a^
	IsiZulu	24.9 ^ab^	8.3 ^ab^	6.4 ^ab^	11.4 ^ab^	27.8 ^ab^	0.14 ^ab^	25.7 ^ab^	1.6 ^ab^	1.1 ^ab^	3.0 ^bc^	170.7 ^abc^
Mean		32.8	11.2	8.6	13.6	37.6	0.18	33.9	2.1	1.3	2.6	196.9
Third	PAN8816	34.2 ^bc^	9.6 ^abcd^	9.9 ^bc^	11.4 ^ab^	31.9 ^abc^	0.15 ^abc^	29.6 ^abc^	1.8 ^abc^	1.2 ^ab^	2.2 ^abc^	182.4 ^abc^
	Macia	33.9 ^bc^	13.2 ^bcd^	15.8 ^de^	15.7 ^b^	38.6 ^bcd^	0.19 ^abc^	37.7 ^bc^	2.0 ^abc^	1.4 ^ab^	2.7 ^abc^	233.0 ^bc^
	Ujiba	40.8 ^c^	10.1 ^abcd^	17.5 ^e^	16.5 ^b^	37.9 ^abcd^	0.19 ^abc^	36.9 ^bc^	2.4 ^bc^	1.5 ^b^	5.1 ^d^	174.9 ^abc^
	IsiZulu	34.3 ^bc^	9.4 ^abcd^	12.6 ^cde^	15.1 ^ab^	39.4 ^bcd^	0.17 ^abc^	34.3 ^abc^	2.0 ^abc^	1.3 ^ab^	3.5 ^c^	184.2 ^abc^
Mean		35.2	10.7	13.9	14.7	37.0	0.18	34.7	2.1	1.3	3.3	195.2
CV (%)		1.7	1.9	2.3	1.1	1.4	1.6	1.6	0.8	1.3	0.5	3.9
*p*-value (S × G)		<0.001	0.057	<0.001	0.07	0.002	0.009	0.034	0.019	0.191	<0.001	<0.001
*p*-value (genotype (G))		<0.001	<0.001	<0.001	<0.001	<0.001	<0.001	<0.001	<0.001	<0.001	<0.001	<0.001
*p*-value (season (S))		0.056	0.235	<0.001	0.093	0.219	0.710	0.345	0.876	0.117	<0.001	0.022

CV = coefficient of variation; ^abcd^ = means sharing a common letter(s) are not significantly different from each other, while means not sharing a similar letter(s) are statistically different.

## Data Availability

The datasets generated during and/or analysed during the current study are available from the corresponding author on reasonable request.

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
