# Peer review of "Assessing Suitability of Sorghum to Alleviate Sub-Saharan Nutritional Deficiencies through the Nutritional Water Productivity Index in Semi-Arid Regions"

_foods, 2021, doi:10.3390/foods10020385_

Round 1

Reviewer 1 Report

Dear Author,

I reviewed your manuscript entitled “Assessing suitability of sorghum to alleviate Sub-Saharan nutritional deficiencies through the nutritional water productivity index in semi-arid 
regions.”.  

The submission falls inside one of the scopes of Foods (food chemistry and physical properties but, unfortunately, inadequate description of climate and agronomic practices coupled with unsuitable statistical analysis and discussion make results not sufficiently far-reaching.

The methodology is stimulating, even if NWP is already known, more info about sorghum should be fascinating for local scopes.

The main limitation of the paper is the layout which is not clearly addressed and is limited to a single season without any description of climate. Even if a reference is given for details, presently it seems like a report on some additional chemical analysis, but specific responses seem to be related with local conditions (and referred to local genotypes and to a single year!). Consequently, conclusive assessments about nutritional water productivity cannot be clearly drawn, as it would be expected from a paper published in an international journal like Foods.  

Some results are confusing both because statistical analysis is not clear and because sentences that are not clear and concise.

Sincerely

General issues

English should be checked; it is generally correct but sometimes some verbs are missed and some words are not correct and some mistakes are present (only eg line 52 research have line 60 to addressing)

Latin name should be added at the first citation (abstract).

Some inaccuracies are present lowering the quality of the paper.

Introduction

It is verbose and with too many repetitions. E.g. twin issues of food and nutritional…is repeated many times. State the problem once for all and then eventually describe how evaluation on water efficiency for protein/starch mineral unit is important and what is known for sorghum and what remains unknown.

M&M

Many pieces of information are missed (even if a reference is given for details).

Description of methods is inadequate to guarantee reproducibility.

Agronomic management of the crop is missed and the same is true for meteorological and soil characterization.

One year of experimentation in open field is not demonstrative and results cannot be generalized.

I would suggest addressing results to a trial on planting dates’ effects on the water efficiency of some nutritional characteristics.

Why those specific genotypes were tested? Were the genotypes differing in cycle length? But, more important, why these genotypes were evaluated? Which was the hypothesis behind their choice?

Nutrition guidelines consider energy, protein and fat and 10 mineral components. Why in the present research those specific characters were determined? This should be discussed.

Results

They are interesting even if not so novel and related to a particular environment (and year) and local genotypes.

They should be better endorsed by statistical analysis and consequently properly reported (and discussed).

Presently, results are difficult to understand, both for English mistakes both for illogical sequence.  
Sometimes they don’t look endorsed by statistical analysis. For example, tables are not clear, as they are not all reporting an interaction,

May be that highlighting, at the beginning, which treatment is significant could be helpful in understanding main differences among treatments.  Insufficient statistical analysis that makes the most interesting results (about the possible interactions) not endorsed;

Anyway, I tried to check the significance as reported in table 4 and results presentation is not in accordance with the reported LSD (13.9 for starch as an example).

If in the introduction there is a nice explanation of what is known and what remains to be studied in sorghum NWE, then discussion should address these topics and explain why some traits have been found to be affected by genotypes and/or planting dates and their interactions.

Author Response

All comments have been attended to, and author comments to suggested revisions are contained in the uploaded document.  

Reviewer 2 Report

#Manuscript: Assessing suitability of sorghum to alleviate Sub-2
Saharan nutritional deficiencies through the nutritional water productivity index in semi-arid regions.

Generally, the text of the manuscript is organized and well-written academically. Methods applied are appropiate. Moreover, valuable data are reported and references are enough to discuss the results presented. 

Some minor points should be addressed before publishing:

L84: Could the authors give any detail of the experimental design? In my opinion it is an aspect important for the reader and the give the reference of this, it is not appropiate. 

L94-97. This is an objective. And is more easy to read that the L69-72.

Table 1: Please, add a footnote with the meaning of initial, development, and mid-season. 

L118 Any reference o mineral composition determination?

L130: And the results were expressed in...? Add this, please

L143. At confidential level of 95%.

Table 2,3,4. Add a footnote with the meaning of LSD and CV(%)

The conclusions section is a little bit messy. L294-2948 should be ommitted (repetitive information). Please, summarize this section just with the main results obtained in the work. Do not included references in this section [35]. 

Author Response

All suggested revisions have been attended to, and author comments to suggested revisions are contained in the uploaded document.

Round 2

Reviewer 1 Report

Dear Authors,

The manuscript has been deeply improved and can now be accepted.

Sincerely

Author Response

All reviewer comments have been addressed and highlighted in red font color for ease of tracking.